

# Multi-year pair-bonding in Murray cod (*Maccullochella peelii*)

Alan J. Couch, Fiona Dyer and Mark Lintermans

Centre for Applied Water Science, University of Canberra, Canberra, ACT, Australia

## ABSTRACT

Mating strategies in fishes are known to include polygyny, polyandry and monogamy and provide valuable insights regarding powerful evolutionary forces such as sexual selection. Monogamy is a complex of mating systems that has been relatively neglected. Previous work on mating strategies in fishes has often been based on observation and focused on marine species rather than freshwater fishes. SNPs are increasingly being used as a molecular ecology tool in non-model organisms, and methods of probabilistic genetic analysis of such datasets are becoming available for use in the absence of parental genotypes. This approach can be used to infer mating strategies. The long-term pair bonding seen in mammals, reptiles and birds has not been recorded in freshwater fishes—in every other respect an extremely diverse group. This study shows that multi-year pair bonding occurs in an Australian Percichthyid fish that exhibits paternal care of eggs and larvae. Using SNPs, full sibling pairs of larvae were found over multiple years in a three-year study. Stable isotope signatures of the larvae support the genetic inference that full sibling pairs shared a common mother, the ultimate source of that isotopic signature during oogenesis. Spatial and temporal clustering also suggests that the full sibling larvae are unlikely to be false positive identifications of the probabilistic identification of siblings. For the first time, we show multi-year pair bonding in a wild freshwater fish. This will have important conservation and management implications for the species. This approach could provide insights into many behavioural, ecological and evolutionary questions, particularly if this is not a unique case. Our findings are likely to initiate interest in seeking more examples of monogamy and alternative mating strategies in freshwater fishes, particularly if others improve methods of analysis of SNP data for identification of siblings in the absence of parental genotypes.

## INTRODUCTION

Pair-bonding is widely documented among vertebrates and is reported in mammals, birds, reptiles, and fish. Pair-bonding is often associated with monogamy, site fidelity, shared parental care and a strong affinity between individuals (*De Waal & Gavrilets, 2013*), but none of these characteristics is exclusive to pair-bonded animals. Pair-bonding may be short, medium, long-term or even lifelong. Monogamy is a complex of mating systems that has been relatively neglected (*Mock & Fujioka, 1990*). Fish are under-represented in the pair-bonding literature generally, but there is little reason to suppose it does not occur as a successful mating strategy in the fishes with their diverse and ancient evolutionary lineages.

Corresponding author
Alan J. Couch,
alan.couch@canberra.edu.au

There are examples of pair-bonding in marine species such as the seahorse (*Hippocampus whitei* Bleeker, 1855) (*Vincent & Sadler, 1995*, *Hippocampus subelongatus Castelnau, 1873*) (*Kvarnemo et al., 2000*) the French angelfish (*Pomacanthus paru* Bloch, 1787) (*Whiteman & Côte, 2004*), hawkfish (*Donaldson, 1989*) and the Ceratiidae family of Lophiiformes (Anglerfish) (*Turner, 1986*). Although genetic monogamy is thought to be uncommon in fish (*Tatarenkov et al., 2006*) it has been seen in bonnethead sharks (*Sphyrna tiburo* Linnaeus, 1758) (*Chapman et al., 2004*) and cichlids (*Steinwender, Koblmüller & Sefc, 2012*; *Takahashi & Ochi, 2012*).

Evidence of other mating strategies in fish is common. Polyandry is reported in sandbar (*Carcharhinus plumbeus* Nardo, 1827), bignose (*Carcharhinus altimus* Springer, 1950) and Galapagos sharks (*Carcharhinus galapagensis* Snodgrass and Heller, 1905) (*Daly-Engel et al., 2006*) and Teleosts such as the lingcod (*Ophiodon elongatus* Girard, 1854) (*King & Withler, 2005*). Harem polygyny is known in an obligate coral-dwelling fish, the pygmy coral croucher (*Caracanthus unipinna* Gray, 1831) (*Wong, Munday & Jones, 2005*). Channel catfish (*Ictalurus punctatus* Rafinesque, 1818) provide a rare suspected example of genetic monogamy in a fish species with uniparental offspring care (*Tatarenkov et al., 2006*).

There are six hypotheses suggested for the evolution of monogamous pair-bonds (*Whiteman & Côte, 2004*). These authors show that paternal care may act to increase the likelihood of monogamy in combination with each of the proposed hypotheses through decreased benefits to males from searching for additional mates or increased advantages to females from sequestering a single high-quality mate (*Whiteman & Côte, 2004*). Other researchers also argue that monogamy results from the need to guarantee a high-quality mate and territory in a competitive environment (*Morley & Balshine, 2002*).

Fish species have highly diverse breeding behaviors that make them valuable for testing theories on genetic mating systems and reproductive tactics (*Avise, 2002*) but medium, long or life-long pair-bonding in freshwater species in the wild has not been reported in the literature. One example of short-term pair-bond has been demonstrated experimentally in a mouthbrooding cichlid (*Xenotilapia rotundiventralis* Takahashi, Yanagisawa & Nakaya, 1997). This is demonstrable because of its genetic monogamy and parental care requiring the transfer of embryos from the female to the male after three days (*Takahashi & Ochi, 2012*). Similarly, *De Woody et al. (2000)* showed monogamy in largemouth bass (*Micropterus salmoides* Lacepède, 1802). Another example of monogamy is seen in *Neolamprologus pulcher* Trewavas & Poll, 1952 (*Desjardins et al., 2008*). These genetically identified monogamous pairings each were for a single breeding event. There is no obvious intrinsic reason for long-term pair-bonding to be underrepresented in the repertoire of freshwater fishes mating systems.

The absence of identification of long-term pair-bonding in freshwater species may have a number of explanations. For example, there may have been less research effort compared with marine systems, perhaps because making observational studies in turbid or high-energy freshwater systems is very difficult. Tagging or radio-tracking adult fish in freshwater systems could produce circumstantial evidence of pair-bonding, but even this sort of long-term or periodic co-location evidence has not been reported. Modern genetic techniques, particularly genome reduction methods with a large number of markers

available, and the development of bioinformatics methods to analyse these larger data sets, along with the relevant spatial and temporal data, now allows for cost-effective identification of genetic monogamy and pair-bonding (*Macdonald et al., 2005*; *Razick, 2016*; *Bayerl et al., 2018*).

### Identifying pair-bonds in an Australian Freshwater Fish

The iconic Australian Murray cod *(Maccullochella peelii)* (Mitchell, 1938) is one of four morphologically cryptic but genetically distinct species within *Maccullochella* (*Rowland, 1993*; *Nock et al., 2010*). The Murray cod is Australia's largest freshwater fish and can grow to as large as 180 cm in length with a maximum recorded weight of 113.6 kg. Its fecundity ranges from 9,000–120,000 eggs annually. The species is highly sought after by anglers. It is limited to parts of the Murray-Darling Basin (MDB) and is listed as vulnerable under the Environment Protection and Biodiversity Conservation Act (1999) (*Australian Department of Agriculture, Water and the Environment, 2020*). Murray cod still form an important recreational fishery (*Lintermans & Phillips, 2005*; *Koehn & Todd, 2012*; *Ye et al., 2016*).

Murray cod is a species worth studying for potential evidence of long-term pair-bonding and alternative mating strategies. Male Murray cod are known to provide parental care to both eggs and larvae for up to 20 days after preparing a nest area. They are long-lived (up to 48 years), slow-maturing (*Lintermans, 2007*) and show high site fidelity (*Koehn & Nicol, 2016*). These are life history factors associated with long-term pair-bonding in other species *Kleiman, 1981*; *Barlow, 1988*; *Arnold & Owens, 1998*; *Hatchwell & Komdeur, 2000*; *Chapple, 2003*. Murray cod have been shown to exhibit monogamous mating, as well as polygyny and polyandry over three breeding seasons when held captive in ponds (*Rourke et al., 2009*).

In this study single nucleotide polymorphisms (SNPS) were used to look for sibling relationships of Murray cod larvae and these data, in combination with relevant metadata, were used to infer the existence of multi-year pair-bonding as a mating strategy. The probability of multi-year pair-bonding occurring is considered. Although SNPS are increasingly being used to identify parentage (*Vandeputte & Haffray, 2014*; *Huisman, 2017*) confidence in the methods is still developing. For this reason stable isotope evidence is provided to support the genetic findings. This reduces the possibility that any relatedness found was not due to contamination during sample processing, or is merely an artefact of the probabilistic approach to inferring sibship relationships in the absence of known parentals.

## METHODS

This study combines spatial and temporal data from three years of larval sampling with genomic data (SNPS) to infer relationships between larval Murray cod. Carbon and Nitrogen stable isotope data is used to validate the relationship data that has been generated.

### Study site

We sampled Murray cod larvae from six sites along a 50 km upland reach of the in Murrumbidgee River in the Australian Capital Territory (ACT), Australia (Fig. 1).

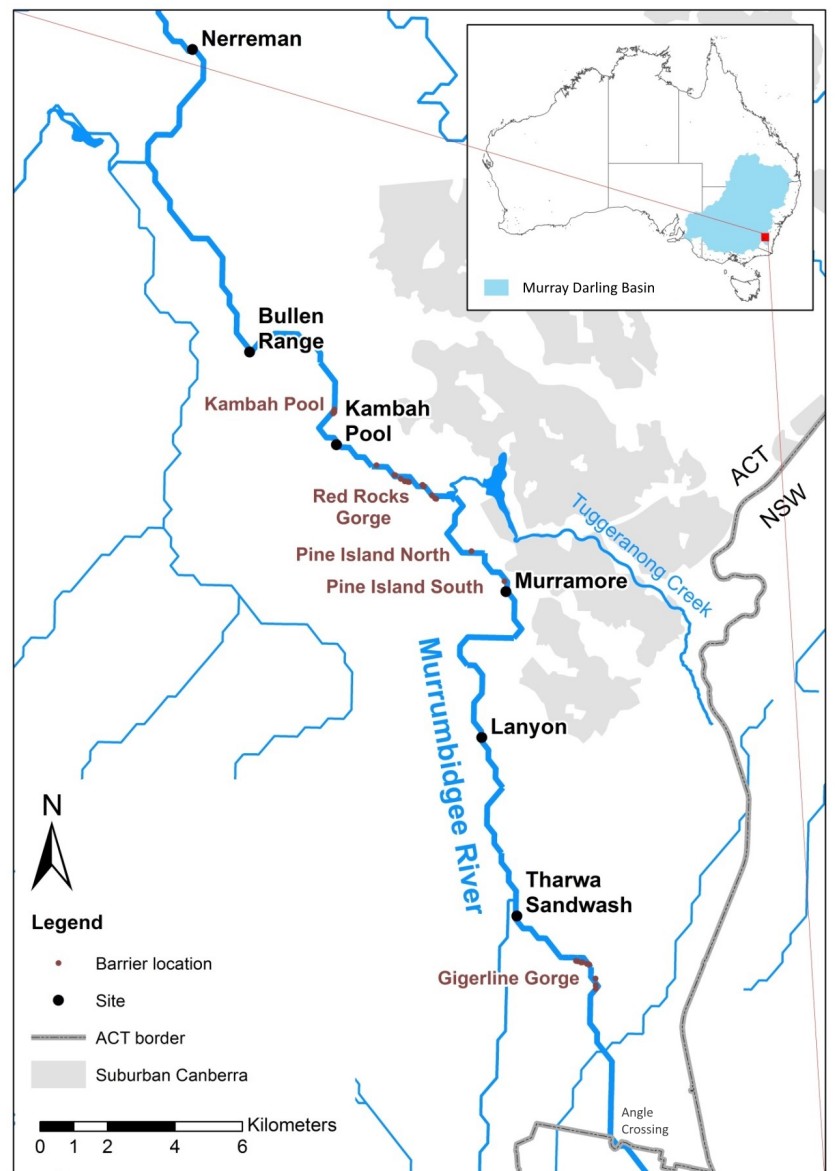

**Figure 1** **Upper Murrumbidgee river study area.** Collection sites are shown in bold black text and putative barriers to adult fish migration are shown in red text. Barriers from F Dyer, M Lintermans & A Couch (2014, unpublished data).

### Larval fish collection, identification, preparation and aging

Data were collected as previously described in *Couch et al. (2016)*. Specifically we examined 261 of 2,607 *Maccullochella* larvae which were collected in 2011, 2012, and 2013 from the six sites (Fig. 1) using larval driftnets. The sympatric *Maccullochella* species Trout cod (*M. macquariensis*) and hybrid larvae were identified using a combination of SNPs and mitochondrial DNA sequencing (*Couch et al., 2016*). These were excluded from the dataset leaving 251 larval Murray cod used for this analysis. Larvae and tissue samples were
preserved in 95% ethanol at room temperature until 2014 after which samples were stored at −20 °C. Fish were collected under ACT Government licences LT2011516, LT2012590 and LT20133653. The research was conducted under approvals CEAE 11-15 and CEAE 13-17 from the University of Canberra Committee for Ethics in Animal Experimentation.

### Age of larval fish

Age estimation has previously been based on larval length (*Koehn & Harrington, 2006*), larval otolith diameter (*Vogel, 2003*) and daily growth increments in larval Murray cod otoliths (*Humphries, 2005*). In the present study, age was estimated using a combination of the above methods. Estimates were made based on otolith size using the mean sagittal otolith length of both otoliths where possible for each larva. While this is less accurate than daily increment ageing it does allow many more larvae to be aged in the time available. Mean otolith lengths of both sagittal otoliths were calculated for 365 larvae. Subsets of 29 of 84, 29 of 51 and 31 of 230 were aged by a commercial provider for the years 2011, 2012, 2013 respectively. From this, a curve was developed for each year and estimated ages, based on mean otolith length, was calculated (*Couch, 2018*).

### Estimation of spawning and hatch times

Murray cod larva hatch from eggs deposited on hard substrates after 4.5–13 days (depending on water temperature), (*Koehn & O'Connor, 1990*; *Koehn & Harrington, 2005*; *Koehn & Harrington, 2006*) or 3–8 days (*Humphries, 2005*), and drift in the water column for some time (*Humphries, 2005*). In this study, the spawning to hatching time (Incubation) which is known to be a temperature dependent process was estimated using the formula developed by *Ryan et al. (2001)*. That is:

$$\text{Incubation}_{(duration)} = 20.67 - 0.667 * [\text{Water Temp}(°C)] \tag{1}$$

The median of the estimates of the duration of brood care (4–10 days) (*Humphries, 2005*) and dispersal (4 days) (*Gilligan & Schiller, 2003*) were, with larval age, used to back-calculate spawning and hatch dates, as day-of-year (DoY), and from that, dispersal duration. The process, following spawning migration and courtship, can be summarised as:

**Spawning** |Incubation(egg care)|**Hatch**|Brood-Care|Dispersing-**Capture|**

Therefore:

$$\text{DoY}_{(Hatch)} = \text{DoY}_{(Capture)} - \text{Age} \tag{2}$$

$$\text{DoY}_{(Spawning)} = \text{DoY}_{(Hatch)} - \text{Duration}_{(Incubation)} \tag{3}$$

$$\text{Dispersal}_{(duration)} = \text{Age} - \text{Broodcare}. \tag{4}$$

Once dispersal, age and capture parameters were known, these equations were then used to calculate hatch and spawning dates.
## Putative nest location

Individual nest sites are unknown so an estimate of the putative nest site was made based on a mean larval dispersal velocity of 700 m per day for the duration of the time available for dispersal based on the larva age (*Couch, 2018*).

## Genomic DNA extraction and sequencing

Genomic DNA Extraction and Sequencing was performed as previously described in *Couch et al. (2016)*. Total DNA was isolated from whole larval heads. The DNA extraction protocol is detailed in *Couch & Young (2016)* and is based on a turtle DNA extraction protocol (*FitzSimmons, Moritz & Moore, 1995*). Sequencing was done using Diversity Arrays Technology's (DArT) DArTseq$^{TM}$ which represents a combination of DArT complexity reduction methods and next-generation sequencing platforms (*Kilian et al., 2012*; *Courtois et al., 2013*; *Cruz, Kilian & Dierig, 2013*; *Raman et al., 2014*). Sequences generated were processed using proprietary DArT analytical pipelines (http://www.diversityarrays.com).

## Marker scoring and statistical analysis

Marker Scoring and Statistical Analysis was performed as previously described in *Couch et al. (2016)*. Specifically, DArTsoft (Diversity Arrays Technology, Building 3, University of Canberra, Australia), a software package developed by DArT PL (http://www.diversityarrays.com/software.html), was used to both identify and score the markers that were polymorphic.

### SNP analyses

SNP Analyses have been described in *Couch et al. (2016)*. Variation in the genome-wide SNP data of the studied Maccullochella genotypes was analysed using Discriminant Analysis of Principal Components (DAPC) using sequential K-means and model selection to infer genetic clusters (*Jombart, Devillard & Balloux, 2010*) using R package 'adegenet'version 2.0.1 (*Jombart, 2008*). The data were converted into a genlight object (the format required by the software) and three principal components were retained. Two principal components were plotted using ggplot2 version 2.1 (*Wickham, 2009*). Summary and comparative statistics were created in R version 3.3.0 (*R Development Core Team., R Core Team, 2013*) and Tableau version 9.2 (*Tableau, 2013*). Maps were created using ArcGIS version 10 (*ESRI, 2013*) and Tableau.

## Carbon and nitrogen stable isotope analysis

Dried muscle material from each fish larva (0.86 mg $\pm$ 0.17 SD), the bulk of the posterior portion of its body without head and gut of the fish, were encapsulated in tin. Samples were combusted in an elemental analyser mass spectrometer (Sercon, Crewe, United Kingdom) at the Australian National University Research School of Earth Sciences Radiocarbon Laboratory, on a fee-for-service basis and assayed for $\delta^{15}N$ and $\delta^{13}C$ stable isotope ratios and C:N ratio. Isotopic signatures were determined based on Australian National University isotopic standards (USGU41, USGU40, Caffeine and Gelatine). Measurement precision was approximately 0.08 ‰ for 13C and 0.15 ‰ for 15N. Isotope values are expressed as the relative parts per thousand (‰).

### Inferring existence of family groups using spatio-temporal data

Plots were made using hatch date and putative nest location for larvae from each annual cohort. Individual clusters, corresponding to putative nests, were identified and nominated.

### Inferring sibship using related

The probability of relatedness (r) was calculated for individual larval dyads based on the trioml algorithm using 'related' (*Pew et al., 2015*).

A simulation was run to identify a probability density function for full siblings, parent–offspring, half-siblings, and unrelated individuals using the allele frequencies from the DARTseqs for each larva. A probability estimate that would best estimate the cut-off probability between full siblings and half-siblings was identified and used to subset likely full-sibling dyads from the larval dyads. Full siblings were assigned a name representing their putative 'mother'. This was arbitrary and could have been assigned a name representing putative 'father'.

The relationships between individuals based on the dyads and the probabilities was visualised by making network graphs and plotting them using Gephi (*Bastian, Heymann & Jacomy, 2009*) and 'r' package 'iGraph'(*Csardi & Nepusz, 2006*). The set of dyads containing full siblings was used to prepare network graphs to facilitate visualisation of the family groups and the assignation of a name to putative female parent.

### Inbreeding coefficient

The two likelihood algorithms - dyadML and trioml - as well as the lynchrd, and ritland algorithms within the 'r' package 'related' can account for inbreeding in their estimates of relatedness. The command "allow.inbreeding=TRUE" was set to output an inbreeding coefficient for each individual under each of the three algorithms above.

### Comparison of inferred family groups and sibships

The plot illustrating family groups previously identified and named using spatio-temporal data alone was then coloured by the name of the putative mother. This comparison increased precision in identification of 'family' groups.

## RESULTS

### Accounting for outbred population

Using the measures of relatedness (r) calculated using the trioml algorithm it became apparent that there were two distinct groups within the larvae sampled; those that were highly outbred (r<−0.4), and the rest which were not strongly inbred or outbred (r>−0.4 and <0.3). Relatedness amongst the non-outbred fish was used to determine common parentals. Principal component analysis of those larvae considered to be outbred suggested a difference between the two populations. These differences were not correlated with location or year. Fish with a coefficient of inbreeding below −0.4 were considered outbreds and separate from the 'river' fish (Figure in supplementary material). These fish were excluded from subsequent related analysis as they were considered likely to be Murray cod introduced from a recent re-stocking program.

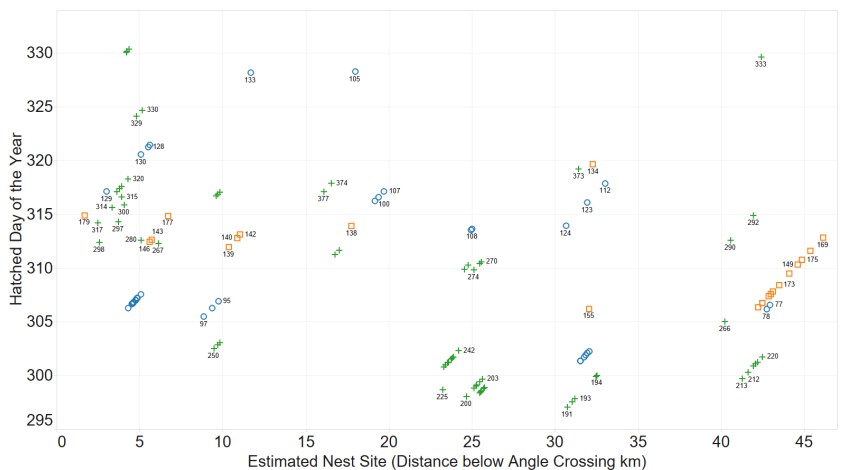

**Figure 2** **Inferring existence of family groups using spatio-temporal data.** Coloured by year: 2011 (blue circle), 2012 (orange square), 2013 (green cross).

### Inferring existence of family groups using spatio-temporal data alone

Initially, the existence of family groups was inferred from a scatterplot of spatial and temporal separation of larvae. Figure 2 shows clusters of larvae distributed over space and time.

### Nomination of putative female parent

The set of dyads containing full siblings was used to prepare network graphs to facilitate visualisation of the family groups and the assignation of a name to putative female parent. Four female Murray cod mated with the same male for more than one year of the three years sampled (Fig. 3). One pair mated for three years sequentially and three other pairs mated for two of the three years studied, one pair detected mating sequentially and two pairs detected mating in non-sequential years.

### Comparison of inferred family groups and sibships

A spatio-temporal plot was coloured by the putative female parent of the larvae (Fig. 4) and it is a clear correlation between identification of groups by spatio-temporal factors alone and those identified by genetic relatedness of the nest groups.

When the Carbon and Nitrogen Isotope ratios of the larvae are considered the strong clustering of CN isotope ratios and female parent illustrates a high correlation between full sibling status and body isotope signature provided to the larva from its female parent (Fig. 5). This also supports the validity of parentage assignment based on related analysis.

The identification of sibling relationship is also supported by the limited number of observations (just 3 individuals of 35 full sibling pairs) that were found at separate sites, and those three were only at detected at sites immediately adjacent to the other sibling. These larvae sibling pairs are 140,179; 133,105 and 170,191 can be seen in Fig. 4.
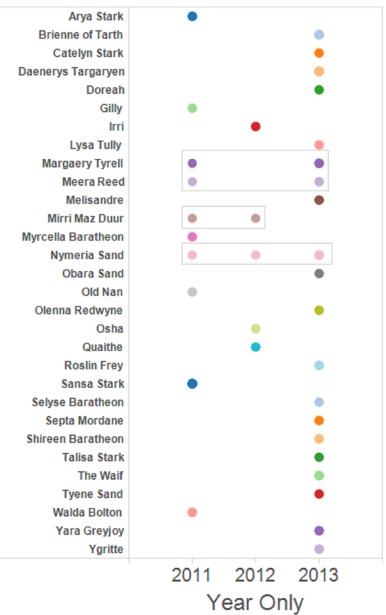

**Figure 3 Full sibling dyads were identified and assigned a putative female parent name.** Full sibling larvae from multiple years are boxed. The rest are full sibling pairs found only within that one year. Singleton larvae—those without any identified full siblings—are not shown.

## Inferring sibship using related

Simulation using the allele frequencies present in the population produced the probability density function in Fig. 6. In this way, the overlap between relatedness values can be assessed. In this case, a 'cutoff' value of any r value above 0.4 was selected to identify most full sibling dyads while minimising the possibility of inadvertently misclassifying half-siblings as full siblings (Fig. 6). The possibility of parent–offspring relationships is obviated because all larvae in the analysis were collected within three years and the sexual maturity of Murray cod is greater than 4–5 years (*Lintermans, 2007*).

After identifying the optimal value of r to eliminate dyads least likely to represent full siblings, the set of dyads were filtered to include full siblings only. This resulted in 35 dyads (pairs of full siblings) that were assigned to a family group.

## Probability of observed multi-year pair-bonding

The multi-year bonding identified in this case is unlikely to be due to chance alone. The probability of multi-year pair-bonding occurring within the cohorts by chance can be estimated if:

y = number years (2 or 3) with the same mate; and

n = number of pairs (138) including singletons (larvae with no siblings found).

and we assume 50/50 sex ratio in accordance with the findings of previous research on the species (*Cadwallader, 1977*; *Koehn & O'Connor, 1990*) then the probability of pairing is:

$$p = (1/n - 1)^y - 1. \tag{5}$$

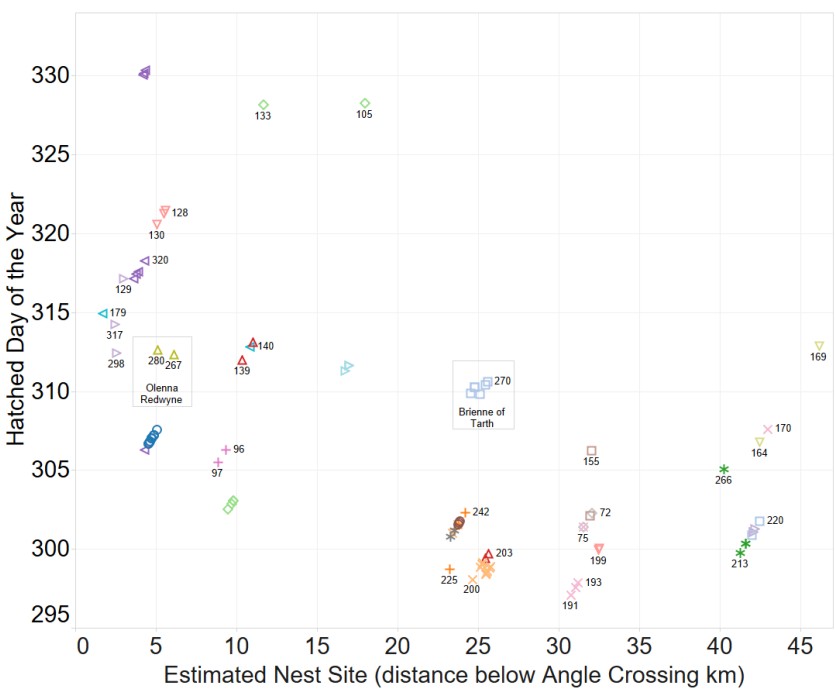

**Figure 4   Inferring existence of family groups using spatio-temporal data coloured by the nominal female parent.** Some larval labels and two mother's names are shown for reference. Larval sibling pairs collected from adjacent sites are 140,179; 133,105 and 170,191.

Applying Eq. (5) for an individual female, the probability of mating with the same male for two years is <0.008 and for each individual female the probability of mating with the same male for three years is <0.00005. Clearly, the probability of four individuals each choosing the same mate for multiple years reduces this probability even further.

To put it another way; for p to even approach non-significance at the 0.05 level, the number of available males would need to be as low as five. Thus it seems unlikely to be random mate selection.

## DISCUSSION

This study has, for the first time in the wild, allowed us to infer that some male and female Murray cod pair-bond for more than one year at a time. This suggests that Murray cod exhibit long-term pair-bonding under some conditions. Our study does not provide any evidence that polyandry or polygyny are absent, nor does it provide evidence for or against the coexistence of alternative mating strategies. Data similarly derived on half-siblings status may be able to provide significantly more detail of mating strategies in use. Nevertheless, if pair bonding is a feature of long-lived freshwater fish, then it has profound implications for management and conservation strategies.

Because of the limited numbers of full sibling pairs detected across years (4/35 pairs in three years), it might be concluded from these data that multi-year pair-bonding is not a commonly adopted mating strategy, and this may be the case. However, such a conclusion

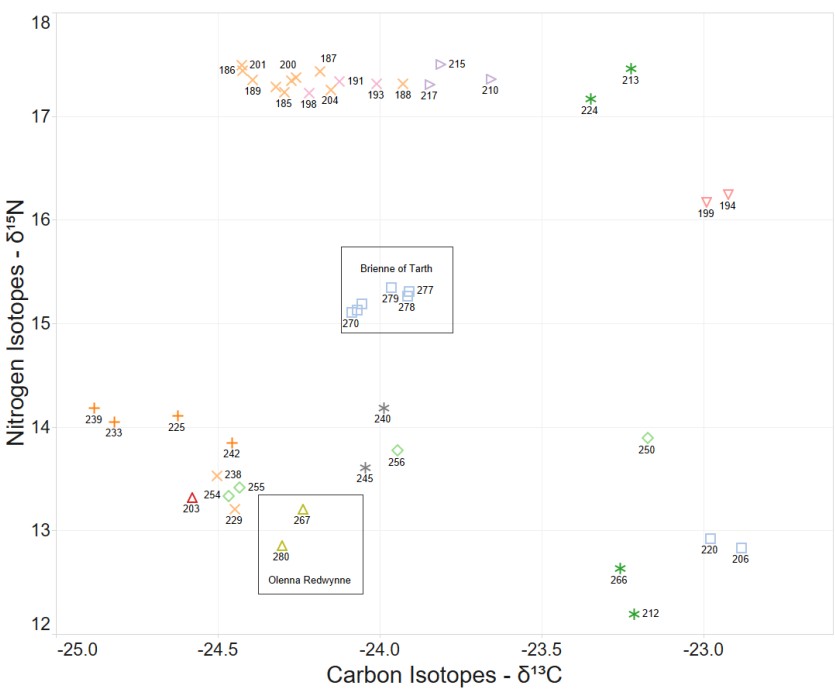

**Figure 5** *δ15N and δ13C stable isotope ratio plot coloured by nominal female parent.* Some larval labels and two mother's names are shown for comparison with Figure 4.

is considered unwarranted because the spatial and temporal distribution of sampling of the current study reduces the possibility of detecting full siblings by its limited resolution. Furthermore, recreational fishing pressure may reduce the likelihood of identifying multi-year pair-bonds by eliminating some adult fish from the breeding pool each year. Murray cod are an iconic Australian recreational target and are, if stocking numbers are a guide, the second most sought-after native freshwater fish in the Murray-Darling Basin (*Reynoldson, 2017*) and while recreational fishing regulations prohibit the take of Murray cod during the spawning season (*NSW Department of Primary Industries, 2017*) catch and release is not prohibited, with the aggressive behaviour of adults during this time well known. The risk of being removed from the seasons breeding cohort even applies to angled fish which are subsequently released, (*Henry & Lyle, 2003*) because such fish have been shown to reabsorb oocytes after the stress of capture and release (*Cooke & Suski, 2005*). In this case, a female may return to her home territory rather than pursue a breeding opportunity. In turn, the male would then be more likely to select another mate for that breeding season. A similar disruption is also likely should it be the male that is caught and pulled from his nest territory before spawning or while nest guarding. Examining more larvae for sibling pairs would help quantify the prevalence multi-year pair-bonding as a mating strategy. So too would conducting a multi-year study in waters closed to fishing.

The probability estimates in this study make some assumptions that need consideration. Firstly, the possibility of all males and females being able to access each other across the study reach was assumed. Movement studies of adult fish elsewhere indicate Murray cod
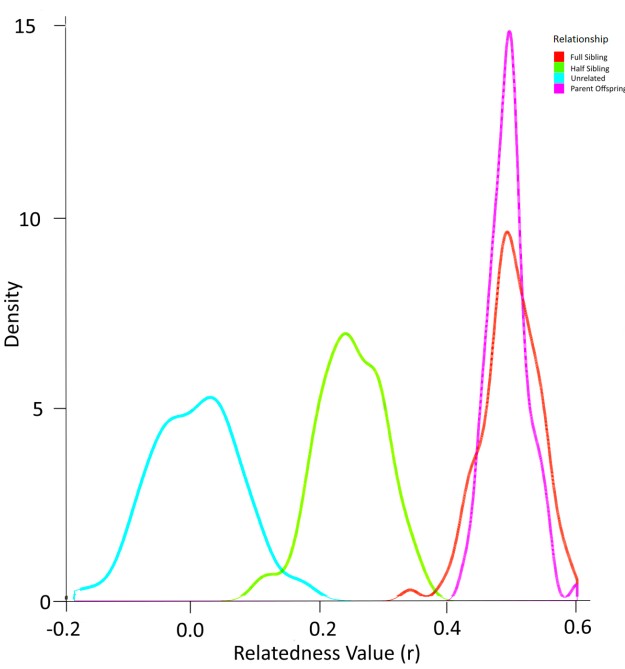

**Figure 6** Probability density function for relatedness (r) as simulated by 'r' package "related" based on the allele frequencies in larval Murray cod 2011–2013 in the Murrumbidgee River.

can and sometimes do undertake long spawning migrations (*Lintermans, 2007*; *Koehn & Nicol, 2016*) and there are fewer than five physical barriers to adult fish movement in this reach when river height peaks and some barriers drown-out most years (F Dyer, M Lintermans & A Couch (2014, unpublished data). While some philopatry is possible in wet years, considerable variation between Murray cod spawning movements has been reported by researchers who hypothesised that some individuals may switch between migratory and sedentary behaviours between years (*Koehn et al., 2009*). What is clear is that further multi-year fine-scale investigations are required on spawning site use. Secondly, we have assumed that strong size-assortative mating preferences have not unduly limited potential mating partners for individual Murray cod. There is no data on size-assortative mating in Murray cod, but it is potentially a factor that may cause two individual fish to come together disproportionately often. This could be because of size selection per se or because larger fish claim the best nesting sites each year. The authors consider very strong philopatric or assortative mating unlikely and in any case the result is still multi-year pair-bonding.

Other physiological and behavioural factors have been associated with monogamy and/or long-term pair-bonding. Some of these may provide avenues for further understanding details of pair-bonding in Murray cod. These include mate recognition (*Sogabe, 2011*), hormones such as oxytocin's role in social bonding (*Acher, Chauvet & Chauvet, 1995*; *Donaldson & Young, 2008*), variation in operational sex ratio (*Sogabe & Yanagisawa, 2007*), sexual dimorphism (*De Waal & Gavrilets, 2013*), and sex role reversal (*Sogabe & Yanagisawa, 2007*).

Multi-year bonding in freshwater fish is a novel and important finding that may change attitudes to these animals and angling. It has been seen in captive Murray cod (*Rourke et al., 2009*) and now evidence of it existing in the wild has implications for decisions of fisheries managers. Efforts to minimise disruption to pair-bond formation may need to reconsider fishing access during the breeding season. This may entail closures rather than prohibition on take which permits catch-and-release. Higher resolution spatial and temporal sampling would allow not only more certainty regarding identified mating strategies employed by the species but also provide valuable data regarding larval dispersal, which still is an important question in Australian freshwater fishes in general and Murray cod in particular.

This study was designed primarily to explore spatial and temporal patterns in larval dispersal in an upland river, and any hybridisation with a recently reintroduced formerly sympatric species. That such an exciting finding as hitherto unknown multi-year pair-bonding was detected in part due to serendipity emphasises the knowledge gaps regarding even some basic life history traits of freshwater fish. These include:

- When are pair-bonds formed? Early on in courtship, or just before spawning,
- Does pair-bonding also occur in reaches where there are no barriers to movement?
- Do bonded pairs co-locate during the non-breeding season?
- What are the impacts of catch and release on long-term pair-bonding? Is it seasonal disruption or more permanent?

## CONCLUSION

Our claim that wild long-term pair-bonding has for the first time been identified within a large group of animals (the freshwater fish of Australia, and perhaps the world) requires more investigation. This study provides a body of evidence—by no means definitive—that such a mating strategy does exist, in at least one freshwater species. This work does not get to the important question of why such a mating strategy may have been adopted by this species. It is however the necessary first step towards such work that may consider this question. Clearly, subsequent work should investigate pair-bonding in Murray cod, and seek to identify pair-bonding in other freshwater fishes in Australia and elsewhere.

### Funding
This work was supported by the Australian Government and ICON Water in the ACT, Australia. The funders had no role in study design, data collection and analysis, decision to publish, or preparation of the manuscript.

### Grant Disclosures
The following grant information was disclosed by the authors:
Australian Government and ICON Water in the ACT, Australia.

### Competing Interests
The authors declare there are no competing interests.

## Author Contributions

- Alan J. Couch conceived and designed the experiments, performed the experiments, analyzed the data, prepared figures and/or tables, authored or reviewed drafts of the paper, and approved the final draft.
- Fiona Dyer and Mark Lintermans conceived and designed the experiments, authored or reviewed drafts of the paper, and approved the final draft, conceived and designed the experiments, authored or reviewed drafts of the paper, and approved the final draft.

## Animal Ethics

The following information was supplied relating to ethical approvals (i.e., approving body and any reference numbers):

The research was conducted under approvals CEAE 11-15 and CEAE 13-17 from the University of Canberra Committee for Ethics in Animal Experimentation.

## Field Study Permissions

The following information was supplied relating to field study approvals (i.e., approving body and any reference numbers):

Fish were collected under ACT Government licences LT2011516, LT2012590 and LT20133653.

## DNA Deposition

The following information was supplied regarding the deposition of DNA sequences:

Data are available at figshare: Couch, Alan (2018): Maccullochella peelii SNPS from Upper Murrumbidgee River. figshare. Dataset. https://doi.org/10.6084/m9.figshare.5383606.v1.

## Data Availability

Collection data are available at figshare: Couch A. 2018. Murray Cod Larval Database Murrumbidgee 2011–2013. figshare. Dataset. DOI: 10.6084/m9.figshare.5715097.v1.

https://figshare.com/articles/Murray_Cod_Larval_Database_Murrumbidgee_2011-2013/5715097.

SNP data as a (genlight object) are available at figshare: Couch A. 2018. Maccullochella peelii SNPS from Upper Murrumbidgee River. figshare. Dataset. DOI: http://doi.org/2010.6084/m9.figshare.5383606.v1.

https://figshare.com/articles/Maccullochella_peelii_SNPS_from_Upper_Murrumbidgee_River/5383606.

## Supplemental Information

Supplemental information for this article can be found online at http://dx.doi.org/10.7717/peerj.10460#supplemental-information.

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
