# Peer review of "Multi-year pair-bonding in Murray cod (Maccullochella peelii)"

_PeerJ, doi:10.7717/peerj.10460_

## Round 0.1 · original submission · Major Revisions

All reviewers and myself are in agreement that the data that you are reporting here is a valuable and interesting contribution to the literature on fish mating systems. However, the reviewers have identified a number of areas that need further attention before the paper is publishable. The reviewers have been very constructive with their comments and I hope that you are able to make the suggested changes.

Reviewer 1 ·

Basic reporting

The study did not clearly specify hypotheses, presumably because it was more of an exploratory study on mating systems in this species. However, given the range of genetic and other variables measured in the stufy in their estimation of mating systems, a set of clearly defined predictions somewhere at the end of the Intro regarding what would be expected under different types of potential mating systems (e.g. monogamy vs polygamy) would have been helpful to process the results that were actually obtained.

I also felt that the Introduction focused too heavily on the biases in literature/studies conducted on freshwater versus marine fishes in terms of mating systems. This study doesn't compare the two. Instead, more effort should be put into describing what IS currently known about the mating systems of freshwater fishes, and then discussing how we can use genetic techniques to indirectly infer mating systems (in addition to directly inferring based on parentage or observations). Currently the Intro lacking in information on genetic techniques employed in the study, and their relative effectiveness.

I also suggest cutting paragraphs on parental care (starting lines 54 and 60), because while technically correct, they are not relevant to the topic of the current study.

Line 67: Neolamprologus pulcher has been shown to be monogamous as well as polygamous, and this has been reported in the literature (Desjardins et al 2006 and Wong et al 2008)

Experimental design

No comment

Validity of the findings

Impact and novelty well discussed especially in light of management of freshwater fish. Data seems robust and well-analysed.
Conclusions regarding type of mating system not overstated and need for further verification is highlighted.

Additional comments

Thank you for your attempt to elucidate a tricky aspect of their reproductive behaviour.

Reviewer 2 ·

Basic reporting

Although this paper is basically well writen, there are many serious flaws in the figure legends, and in many cases it is insufficient to explain.
Fig. 2: It is unknown what each numerical value indicates and what "normal" means.
Fig. 3: It does not exaplain what the difference in color of the point is indicating (I think that it is the year of sampling...).
Fig.6: There is no explanation about the color and numerical value of each plot. I can not find a reson why you made a decision to attach numbers and mather's names only to some plots. How is the explanation of the results of lines 257-259 in the maintext shown in this figure?

Experimental design

I think that you should explain in more detail in the introduction section about the SNPs, and the application of it to parentage analyses.

Validity of the findings

no comment

Additional comments

I listed specific comments below.
1) L30: What does "good" mating strategy mean?
2) L32: Kvarnemo et al (2000) reported monogamy in H. subelongathus, not H. whitei.
3) L39-46 & L54-66: I do not think the information in these paragraphs are necessary for your study.
4) L29 & L72: under-represent or underrepresent
5) L79-82: References are required.
6) L95 All organisms live in competitive environment.
7) L97: Refereces are required.
8) L100 & 107: SNPs?
9) L135-145:More detailed explanation of reproductive biology of this fish is required. Is this fish not only to care eggs but also larvae?
10) L148: Abbreviation "DoY" is not defined before.
11) L188: "15" and "13" should be superscript.
12) L199: DArTseqs?
13) L202 & Fig. 5-7: I cannot find a positve reason for giving a unique name to fish. Something like #1, 2, 3 is enough.
14) L234: Incomplete sentence.
15) L298-299: Incomplete sentence.
16) L314-319: unnecessary discussion, remove.
17) L329-338: uncecessary discusiion, remove or shorten.

Reviewer 3 ·

Basic reporting

Clear, unambiguous, professional English language used throughout:

Although the manuscript is generally well-written, I was missing explanations of abbreviations several times (or they came too late; e.g. lines 100, 148, 162). In a few cases, sentences were very hard to follow, as they were too long and convoluted (e.g. lines 102-104, and 321-323). I also think there is room for more explanations, like “In order to achive this, we did that”.

Intro & background to show context:

It is clear you lack some background knowledge on what is know about mating patterns in freshwater fishes. To improve your introduction, and also to nuance your conclusions, I suggest you read these the publications (and references therein):

Barlow, G. W. (1984). Patterns of monogamy among teleost fishes. Archiv für FischWissenschaft, 35, 75-123.
Barlow, G. W. (1986). A comparison of monogamy among freshwater and coral-reef fishes. In Indo-Pacific Fish Biology (Uyeno, T., Arai, R., Taniuchi, T. & Matsuura, K., eds.). Tokyo: The Ichthyological Society of Japan.
Kvarnemo, C. (2018). Why do some animals mate with one partner rather than many? A review of causes and consequences of monogamy. Biological Reviews. doi: 10.1111/brv.12421

Literature well referenced & relevant: Otherwise, yes!

Structure conforms to PeerJ standards, discipline norm, or improved for clarity: As far as I can judge, yes.

Figures are relevant, high quality, well labelled & described:

Seven figures is a lot. Most figures were informative, but the one I would delete first is Figure 3, as I didn’t see its full value. I read the MS printed in black-and-white. This exposed the fact that almost all figures were impossible to understand, without later checking the colour version on my computer. But even with colour, there is more work to do. Please use bold and plain text for inbred and outbred in figure 2, much thicker lines, with different patterns (dashes, dots, etc) in figure 4. In figure 3 and 5, I don’t think colour is needed (or unclear why it would be). For clarity, in figure 3 and 6, please use kilometers (km) as unit, instead of meters combined with K. Also please explain why the dots line up like they do. Is it an artifact of some sort or a true effect of asynchronous hatching?

Raw data supplied (PeerJ policy): I was unable to access the files. The "figshare" site did not work properly.

Experimental design

Original primary research within Scope of the journal: Yes.

Research question well defined, relevant & meaningful.

Actually, based on what it says in the discussion, the original research question was another, and the authors just stumbled on this one. But I’m fine with that. It’s honest; better than pretending they were looking for this result all along.

It is stated how the research fills an identified knowledge gap.

Yes, to some extent, but I think there are more gaps to be filled by this data set (e.g. see my comment on the introduction & background above). In addition, the authors have chosen to limit their scope to full sibs, filtering out all halfsibs. I think that’s a huge mistake. The study and the paper would be so much stronger if they also looked for halfsibs. If it feels overwhelming, maybe they could limit their search to the individuals they know have bred monogamously for 1, 2 or 3 years, to see if they practice polygyny/polyandry at the same time and/or sequentially.

Rigorous investigation performed to a high technical & ethical standard:

I admit that I’m more at ease with microsatellite data than SNPs, and that the R scripts mentioned are unfamiliar to me. But, as far as I can judge, yes.

Methods described with sufficient detail & information to replicate:

I am missing one important piece of information: How and why were the authors able to identify the maternal genotype, but not the paternal? (And why the funky names?) Have the authors taken tissue samples from females but not males, to match the offspring to, or how does this work? Needs clarification.

Validity of the findings

Impact and novelty not assessed.

The authors do try to sell their results as being the first time monogamy is found in a freshwater species of fish. Since it is not completely true (see Barlow 1984, 1986; Kvarnemo 2018 for examples), and PeerJ doesn't appreciate that kind of sales pitch, it should be tuned down/removed, and the “single example” (line 68) rephrased.

Nevertheless, Barlow’s reviews were published before easy genotyping became available, and too few genetic studies have been done since, so there is a knowledgegap to fill, especially for freshwater species (but marine species as well). Yet, largemouth bass (DeWoody et al. 2000: Proc R Soc B, 267, 2431-2437. DOI 10.1098/rspb.2000.1302), and a number of freshwater cichlids have been studied using genetic markers, but mainly for single breeding events. So, as far as I know, these genetically documented freshwater fish pairbonds lasting three years is likely to be a real novelty.


Negative/inconclusive results accepted: N/A

Meaningful replication encouraged where rationale & benefit to literature is clearly stated: N/A

Data is robust, statistically sound, & controlled: Yes.

Speculation is welcome, but should be identified as such: Some speculations in discussion, but all OK.

Conclusions are well stated, linked to original research question & limited to supporting results: See above. Should be rephrased a little.

Additional comments

Detailed comments:

Abstract: Please mention the fact that Murray cod has paternal care in the abstract, as this is likely to be of interest to future readers of yours, especially in relation to the pattern identified by Barlow between monogamy and parental care in freshwater fishes. I therefore urge you to give more information on what is known about parental care in Murray cod in the main text (line 94).

Abstract (lines 12-15) and in methods (lines 108): Unclear why didn’t you trust your genotypic evidence of monogamy, but used isotopes as well. Please justify/motivate.

Line 21: I suggest you rephrase “particularly if improved methods for analysing SNP data for identification of siblings in the absence of parental genotypes become available.” (replace “if” with “as” or “when”, if more appropriate).

Introduction: I suggest you avoid using capitals for species names, unless its named after a person or geographial place.

Lines 24-25: Feel free to add some invertebrate groups into the animal kingdom as well.

Line 33: Kvarnemo et al. 2000 studied Hippocampus subelongatus, not whitei.

Line 61: “in various species males provide parental care”. In fact, in fish taxa that show care at all, male care is most common – more common than female care or biparental care (Gross & Sargent 1985; or Reynolds et al. 2002):

Gross, M. R. & Sargent, R. C. (1985). The evolution of male and female parental care in fishes. Am. Zool., 25, 807-822.
Reynolds, J. D., Goodwin, N. B. & Freckleton, R. P. (2002). Evolutionary transitions in parental care and live bearing in vertebrates. Phil. Trans. R. Soc. B 357, 269-281.

Lines 62-65: Hard to follow. Probably needs more detail.
Line 81: What do you mean by “meata-data”? Try to be more specific.
Line 95: What do you mean by “evolved in a competitive environment”? As opposed to what?
Line 98: Repeated matings: In what sense? Please be more specific.
Lines 102-104: Eh?
Line 156: 700 m per day: is it really that constant? Must depend on rainfall, right?
Line 178: genlight object =?
Line 197: Pew not in reference list.
Line 202: “putative mother” – or father? How would you know?
Line 220-225: I had no idea r = coefficient of relatedness can have negative values. Is that really correct?. And what is a coefficent of inbreeding?
Line 241: see previous comment on filtering out halfsibs.
Line 272: “size based” is not tested here. Please delete.
Lines 277-280: yes, but better to make one good paper than two half hearted ones.
Line 294: which risk? (pls, try to be more specific)
Line 308-313: Right. Any monogamous pair-bonding will reduce the number of available mates for unmated individuals. Then, of course, assortative mate preferences (size, condition, age, whatever) will add other restrictions on suitable mates. I think it’s likely to find mutual mate choice in long-term monogamous animals, if their fitness depends on some traits of their partner (and compensation by mating with multiple partners isn’t an option). So, some assortative mating is likely, but sure, it doesn’t need to result in very strong assortative mating (for example, see section III-6 in Kvarnemo 2018 for more on this topic).
Line 327: Please rephrase to: “dispersal, which still is an”.
Line 335: Please add “also” after “occur”.

---

## Round 0.2 · Minor Revisions

I have been unable to solicit any reviewers for your manuscript unfortunately, probably due to the current COVID-19 issues that many people are facing. However, this paper does fall within my own area of expertise and I have been through the whole manuscript. Overall you have done a very good job of implementing the suggestions of the earlier referees comments and there are only a few relatively minor issues remaining.

Line 49-52. Need the reference in this line, not just the previous sentence.

Line 325 need a space between ‘Effort’ and ‘to’.

Line 263-272. While there is nothing wrong with the equation here per se, there are two issues that need dealing with. Firstly you have the number of pairs as 138. Can you just expand a little here to clarify what these are. Is the total number of possible pairs given the number of males and females in the population?

Second, and probably more importantly, the logic of this equation is that all individuals are equally likely to pair with one another within and across years. It’s not that clear to me what the spatial distribution is of the study within and across years, however, if males and females are philoptric, then it will mean that at a local level the chance of individuals repeatedly re-pairing will be much higher even with completely random pairing. i.e. if a female only chooses a partner in her small patch of the river, and both males and females are quite sight faithful, then the chances of her pairing with a male repeatedly are much higher, and the model based on her ability to pair with all males in the whole river is quite flawed, because she is unlikely to encounter most of them. That needs to be discussed. If you have data on male and female philopatry or territory size, then it can even be modelled.

---

## Round 0.3 · accepted · Accept

Thanks for responding to those final issues. The paper is now acceptable.